# Diversify and Conquer: Diversity-Centric Data Selection with Iterative Refinement

## Abstract

Finetuning large language models on instruction data is an important step in enriching the knowledge learned during pre-training and improving instruction-following capabilities. As the number of instruction datasets continues to grow, selecting the right data to achieve optimal results becomes increasingly important. In this work, we ask a prominent question: *How can we determine the optimal subset of data for effective training?* While much of the existing research primarily emphasizes local criteria, such as instance quality, for subset selection, we argue that a global approach focused on data diversity is more critical. Our approach utilizes $k$-means clustering to ensure that the selected subset effectively represents the full dataset. We propose an **iterative refinement** method inspired by active learning techniques to resample instances from clusters, with the importance and sampling weight of each cluster being reassessed in every training iteration. This method allows us to reduce the effect of outliers and automatically filter out clusters containing low-quality data. Through extensive evaluation across natural language reasoning, general world knowledge, code and math reasoning tasks, and by fine-tuning models from various families, we observe consistent improvements, achieving a 7% increase over the random selection and a 3.8% improvement over state-of-the-art sampling methods. Our work highlights the significance of diversity-first sampling when finetuning LLMs to enhance performance across a broad array of evaluation tasks. Our code is submitted as supplementary materials.

## 1 Introduction

Large language models are trained on vast amounts of data scraped from the internet, containing a wide range of content qualities. (Penedo et al., 2023; Chen et al., 2023; Laurenccon et al., 2023; Marion et al., 2023). Models develop a broad understanding of language and acquire general knowledge from the unstructured data in this *pretraining* phase (Da et al., 2021; Chang et al., 2024) and align with user intent in the *finetuned* stage using instruction datasets which consists of a more structured format of question and response pairs (Chung et al., 2022; Taori et al., 2023; Li et al., 2023; Üstün et al., 2024). Recent years have seen substantial efforts to create datasets using various manual (Conover et al., 2023; Köpf et al., 2024; Singh et al., 2024) and synthetic (Taori et al., 2023; Wang et al., 2022; Shimabucoro et al., 2024) methods, making it increasingly challenging to determine which dataset is best suited for downstream tasks. A crucial question regarding the **scalability** of finetuning LLMs is: "*what is the optimum subset of data that allows for efficient training and captures aspects of the data relevant to downstream tasks?*"

Instances in a dataset contribute to a model's learning process with varying degrees of impact, affecting the model's performance and generalization (Sorscher et al., 2022; Chen et al., 2022). While recent research has predominantly emphasized *local* features, such as the quality of individual instances for subset selection, we argue that prioritizing a *global* feature —**diversity**—yields greater benefits. When selecting a subset of instances, we manage computational complexity while balancing the trade-off between diversity and representativeness (Zhou et al., 2023), ensuring that the subset captures the underlying data distribution (Ivison et al., 2023; Wang et al., 2024b). Preserving a high level of sample diversity during finetuning is crucial for improving generalization capabilities (Zhang et al., 2024; Yue et al., 2024). Wang et al. (2024b) revealed that using a range of instruction datasets can boost downstream tasks. Wang et al. (2024a) provided a theoretical analysis using determinantal point processes to underscore the significance of diversity in the selection of subsets.

However, ensuring diversity during sampling is difficult, and current methodologies fall short of fully addressing this challenge. Most scoring-based subset selection methods prioritize sample quality and characteristics and subsequently apply a diversity filter (Liu et al., 2023; Xia et al., 2024). Still, since diversity is inherently a global property, addressing it only in the second step limits its effectiveness because it lacks a comprehensive view of the entire collection. This limitation often arises because assessing the data collection globally is computationally expensive (Bukharin & Zhao, 2023).

In this work, we propose a scalable iterative sampling and refinement method to efficiently select a subset of instruction data and maximize the diversity of samples. We iteratively refine the sample selection using early training signals from the fine-tuning model and proceed with continued fine-tuning. With the same training budget, we achieve substantial improvements over fixed sampling approaches and previous state-of-the-art data selection methods. We evaluate the finetuned models on a wide range of tasks, including question answering, math, reasoning, and code, and show consistent improvements over baselines. Overall, our experiments and analyses demonstrate that by sampling a small subset of data, we achieve performance improvements of up to 7% over random selection and 3.8% over the previous sampling methods on a wide variety of tasks. In summary, our contributions are as follows:

- We systematically analyze various clustering and sampling methods and demonstrate that $k$-**means clustering** is particularly effective for selecting an optimal, diverse subset of instruction data, especially when paired with a quality sampling step.

- Our simplest variant, which involves efficiently clustering data points and randomly sampling from each cluster, already achieves performance on par with advanced state-of-the-art sampling techniques, without the need for costly LLM scoring. This supports our hypothesis on the importance of diversity and the representativeness of the sampling process.

- We further propose an **iterative clustering algorithm** that simultaneously combines the learning feedback from the training model and optimizes for diversity based on data distribution for effective instruction tuning. This method outperforms previous approaches on all downstream tasks.

We release the code and the data artifacts used in our experiments to facilitate reproducibility and future research.

## 2 METHODOLOGY

### 2.1 STATIC DATA SELECTION

Given a large and diverse set of instruct data $\mathcal{D} = \{x_1, x_2, \ldots, x_n\}$, we select a subset $\mathcal{D}'$ with budget $b \in \mathbb{N}^+$, where $b = |\mathcal{D}'| \ll |\mathcal{D}|$ and finetune a language model and evaluate a selection of downstream tasks. This subset should be a representative sample of the training data, maintaining high quality and offering a diverse range of examples. We propose to define the problem of sample selection for training data of a language model as a clustering problem with clustering objectives where we want to group similar samples together and separate dissimilar samples into different clusters. We explore various sampling methods to ensure the inclusion of optimal samples from different clusters.

For clustering purposes, we consider two main clustering objectives: $k$-center and $k$-means. Both of these two objectives are metric clustering where we are given a set of points $\mathcal{D}$ with distance metric $d : \mathcal{D} \times \mathcal{D} \to \mathbb{R}_{\geq 0}$ and the goal is to pick a set of centers $\mathcal{C} = \{c_1, \ldots, c_k\} \subseteq \mathcal{D}$ of size at most $k$. For $k$-center, we want to pick $\mathcal{C}$ such that the maximum distance of data points to centers is minimized. More precisely, in $k$-center, we want to minimize

$$\max_{x_i \in \mathcal{D}} d(x_i, \mathcal{C}) \tag{1}$$

where $d(x_i, \mathcal{C}) = min_{c_j \in \mathcal{C}} d(x_i, c_j)$ is the distance of point $i$ to the closest center in $\mathcal{C}$. The $k$-means objective is similar to $k$-center objective but instead of looking at the $l_\infty$ norm of the vector that defines the distance of points to $\mathcal{C}$, we look at the $l_2$ norm of this vector. More precisely, in $k$-means,

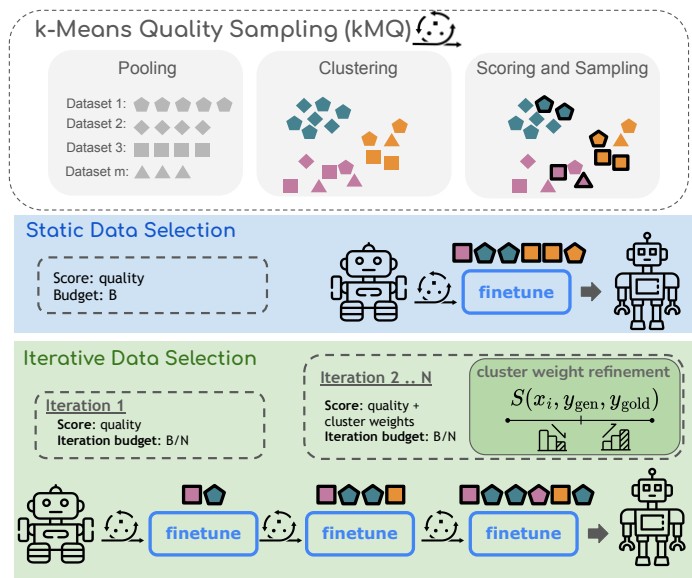

Figure 1: **Our proposed clustering ($k$MQ) and two sampling methods**: We visualize our *static data selection* with $k$MQ, as proposed Section 2.1 and the *iterative data selection* pipeline where we refine the selection criteria and resample new instances in each iteration, as proposed in Section 2.2.

we want to minimize

$$\sum_{x_i \in \mathcal{D}} d^2(x_i, \mathcal{C})$$

Based on this objective and given the set of centers $\mathcal{C} = c_1, \ldots, c_k$, we define $\mathcal{D}_j$ as the subset of data points in $\mathcal{D}$ that are closest to center $c_j$ and belong to the $j^{\text{th}}$ cluster:

$$\mathcal{D}_j = \{x_i \in \mathcal{D} \mid d(x_i, c_j) \leq d(x_i, c_l) \text{ for all } l \neq j, l = 1, \ldots, k\} \quad (2)$$

where $d(x_i, c_j)$ is the distance between data point $x_i$ and center $c_j$.

Beyond the clustering, the next step concerns how to sample data from the clusters with a fixed budget of $m$. We investigate both random sampling and a more informed, quality-based sampling approach. For the quality-based sampling, inspired by the previous approaches (Liu et al., 2023; Bukharin & Zhao, 2023), we propose $k$-means-quality ($k$**MQ**), where we first perform the traditional $k$-means by clustering the instruction data into $k$ centroids, in which $k \ll b$, and sample data from each cluster to form $\mathcal{D}'$. Note that we assign each cluster a budget proportional to its size ($b_j = \frac{|X_j|}{|X|} \cdot b$) and draw samples within each cluster based on the probability weighted by the quality score. We use the same scoring method introduced by Liu et al. (2023) to obtain quality scores, enabling a fair comparison of the hypotheses regarding the importance of diversity-first versus quality-first sampling. More concretely, we sample:

$$\{x_1, x_2, \ldots, x_{b_j}\} \sim \text{Multinomial}(\mathcal{D}_j, \{p(x \mid q)\}_{x \in \mathcal{D}_j}) \quad (3)$$

where $\{x_1, x_2, \ldots, x_{b_j}\}$ is the data sampled from cluster $\mathcal{D}_j$ with replacement, $b_j$ is the budget assigning to the $j^{\text{th}}$ cluster and $p(x \mid q)$ is the probability of picking $x$, weighted by its quality $q$.

Additionally, we take a systematic approach to studying the role of diversity and show the importance of the choice of $k$ in affecting downstream performance, which has been overlooked in previous works (see analysis in Section 4.3).

## 2.2 ITERATIVE DATA SELECTION

---

**Algorithm 1** Iterative Data Selection Pipeline

---

**Require:** $Dataset\ \mathcal{D},\ Budget\ b,\ Iteration\ \mathcal{N},\ base\ model\ \mathcal{F},\ Scorer\ \mathcal{S}$
1: $\mathcal{D}' = \{\}$                                                                $\triangleright$ Selected Data Subset
2: $\mathbf{w}^0 = \{w_0^0, w_1^0, \ldots, w_k^0\} = \underbrace{\{\frac{1}{k}, \frac{1}{k}, \ldots, \frac{1}{k}\}}_{k}$        $\triangleright$ the weights $(w_j)$ of each of $k$ clusters

3: **for** $it = 1$ to $\mathcal{N}$ **do**
4:     $b_{it} = \frac{b}{\mathcal{N}}$                                                    $\triangleright$ Compute iteration budget
5:     $\mathcal{D}' = \mathcal{D}' \cup$ Pick $b_{it}$ from $\mathcal{D}\backslash\mathcal{D}'$ with $\mathbf{w}^{it-1}$         $\triangleright$ Select new subset with budget $b_{it}$
6:     $\mathcal{F}^n = \text{Finetune}(\mathcal{F}, \mathcal{D}')$                             $\triangleright$ Finetune the model for epochs
7:     $\{(x_i, y_{\text{gen}}, y_{\text{gold}})\}_n = \text{Inference}(\mathcal{F}^i, \mathcal{D}')$      $\triangleright$ Generation $J^{it}$ based on the eval instruct
8:     $\mathbf{s} = \{s_1, s_2, \cdots, s_k\}$                                   $\triangleright$ Normalized score for each cluster (Eq. 5)
9:     $\mathbf{w}^{it} = \{w_1^{it}, w_2^{it}, \cdots, w_k^{it}\}$                       $\triangleright$ Adjust selection weight (Eq. 6)
10: **end for**
11: **return** $\mathcal{D}', \mathcal{F}^n$                          $\triangleright$ Return the optimal subset $\mathcal{D}'$ and finetuned model $\mathcal{F}^n$

---

In the previous section, we introduced a two-step approach: sampling a fixed subset of data first and finetuning a model on it. The sampling and finetuning steps are performed independently without any information exchange between the two steps. However, the initial stages of finetuning can offer insights into how individual data points influence the learning process (Anthony et al., 2017; Muldrew et al., 2024). Here, we investigate whether we can improve our sampling method by incorporating early training feedback into the selection process. We accomplish this by periodically increasing the weight of clusters from which the model learns well while decreasing the weight of clusters that the model finds difficult to generalize.

The motivation is twofold: (1) Not all data clusters possess the same level of quality and impact. We further analyze the distribution and quality scores across clusters, revealing significant disparities (see analysis in §4.4). This indicates that some clusters are notably better quality, while others predominantly consist of low-quality data. (2) From a curriculum learning perspective, models can develop different skills and knowledge at varying rates (Xu et al., 2020; Xu & Tewari, 2021; Feng et al., 2023). Increasing the selection from challenging clusters for models to learn can enhance their generalization capability.

Our iterative approach is:

**1. Initialization** Given a fixed training budget of $b$, we use $k$MQ as described in the previous section to cluster and sample an initial set of instances of the size $\frac{b}{\mathcal{N}}$. Next, we finetune the base model for one epoch by going over the sampled data once, using this checkpoint to guide the iterative selection.

**2. Estimation of Sample Difficulty** Using the latest checkpoint, we perform one round of inference on the prompts on which the model is trained. Specifically, given the prompt $x_i$ from the initial sampled set, we generate a new completion $y_i$ from the original seed data, forming the tuple $(x_i, y_{\text{gen}}, y_{\text{gold}}) \in J^i$. We then evaluate the quality difference between $y_{\text{gen}}$ and $y_{\text{gold}}$ using a scorer $\mathcal{S}$. We compute the score for each instance by the following:

$$\mathcal{S}(x_i, y_{\text{gen}}, y_{\text{gold}}) = \text{score}(x_i \oplus y_{\text{gold}}) - \text{score}(x_i \oplus y_{\text{gen}}) \tag{4}$$

where $\oplus$ is the concatenation operator. We explore the effectiveness of different scoring methods in section 4.2.

**3. Resampling** By aggregating and normalizing the scores of samples within each cluster, we modify the sampling weight of each cluster in the next iteration. The goal is to assign a higher weight to the clusters containing higher-quality data while reducing the number of instances selected from lower-quality clusters. We define the score and weight of the $j^{\text{th}}$ cluster as follows:

$$s_j = \frac{1}{|D_j|} \sum_{i=1}^{|\mathcal{D}_j|} \mathcal{S}(x_i, y_{\text{gen}}, y_{\text{gold}}) \tag{5}$$

$$w_j^{it} = \frac{s_j}{\sum_{c=1}^{k} s_c} w_j^{it-1} \tag{6}$$

where $s_j$ is the score of the $j^{\text{th}}$ cluster, $w_j^{it}$ is the weight of the $j^{\text{th}}$ cluster at iteration $it$. $it$ is the iteration number, where $it \in \{0, 1, \dots, \mathcal{N}\}$. $\mathcal{N}$ is the maximum number of iterations and $k$ is the total number of clusters.

We adjust the cluster weights and select $\frac{b}{\mathcal{N}}$ new candidates based on these updated weights. We then train the model and return to step **2**. This process continues until the entire training budget is utilized. The pseudocode summarizing our iterative data selection method is shown in Algorithm 1.

## 3 EXPERIMENTS

### 3.1 TRAINING SETUP

**Source Datasets**   We focus on two large and widely used instruction datasets that include prompts on a diverse set of topics: Alpaca (Taori et al., 2023) and WizardLM (Xu et al., 2023). The Alpaca dataset includes 52K prompts and uses the *self-instruct* framework to evolve seed human instruction datasets and generate a large collection. WizardLM includes 196K prompts where they used *Evol-Instruct* to automatically augment instruction tuning datasets (Alpaca, ShareGPT) to make their instructions more complex (in-depth evolution) and more diverse (in-breadth evolution).

**Encoding data points**   We use Cohere English embedding (`embed-english-v3.0`) to embed the instruction datasets. Note that we encode both the prompts and completions. To study the impact of the embedding model, in Section 4.3 we experiment with other models to encode instances in our training pool, namely OpenAI embedding (`text-embedding-3-large`) and Llama-2-7B model (using the last hidden state of the last token).

| Evalset | Metric | # shots |
|---|---|---|
| MMLU | acc | 5 |
| GSM8k | acc | 5 |
| HellaSwag | acc-norm | 10 |
| ARC | acc-norm | 25 |
| TruthfulQA | acc | 0 |
| HumanEval | pass@10 | 0 |

Table 1: Detailed information of our evaluation settings. For each evaluation dataset, we present the number of few-shot examples and metric adopted for evaluation.

**Training Recipes**   For all experiments, we finetune the `llama-2-7B` base model (Touvron et al., 2023). We train for 3 epochs to achieve convergence and optimal instruction-following performance. We use an AdamW optimizer (Loshchilov & Hutter, 2017), with a learning rate of 1e-5 and 1,000 warming-up steps. The maximum token size is 2048, and the effective batch size is 64. Additionally, in section 4.5 we study the transferability of our findings to other base models and experiment with fine-tuning Mistral (Jiang et al., 2023) and Llama-3-8B (Dubey et al., 2024).

### 3.2 EVALUATION SETUP

To present a comprehensive overview of the performance of our method, we conduct a comprehensive evaluation of our approaches and the established baselines across a range of LLM benchmarks.

**Natural Language Reasoning**   We use HellaSwag (Zellers et al., 2019), and TruthfulQA (Lin et al., 2022). HellaSwag is a test of commonsense inference. TruthfulQA measures a model's propensity to reproduce falsehoods.

**World Knowledge**   We evaluate on MMLU (Hendrycks et al., 2021) and ARC (Clark et al., 2018). MMLU consists of a range of multiple-choice academic questions. ARC is a set of grade-school science questions.

**Code Generation** We use the extensively utilized HumanEval (Chen et al., 2021) benchmark consisting of 164 coding problems to evaluate LLMs' code-writing capabilities at the function level by reporting the pass@10 metric.

**Math Reasoning** We use GSM8k (Cobbe et al., 2021) to evaluate the mathematical abilities of models; GSM8k contains 1319 grade school math test data. We adopt 8-shot testing and report the exact matching.

### 3.3 BASELINES

We implement two strong data selection methods, Deita (Liu et al., 2023) and QDIT (Bukharin & Zhao, 2023) and compare our methods against them. Additionally, we explore other clustering and sampling methods: $k$-center clustering ($k$-Center), where $k$ equals the number of data points, $k$-means-closest ($k$M-Closest), which selects samples based on the closest distance, and $k$-means-random ($k$M-Random), which selects randomly from each cluster, both with the same budget as our proposed approach $k$MQ. We also compare our methods to the random selection of data points.

| | MMLU | GSM8K | HellaSwag | ARC | TruthfulQA | HumanEval | **Avg.** |
|---|---|---|---|---|---|---|---|
| Random | 42.4 | 13.3 | 79.9 | 53.6 | 44.8 | 28.5 | 43.8 |
| Deita (Liu et al., 2023) | 44.1 | 15.6 | 80.1 | 54.3 | 44.9 | 30.8 | 45.0 |
| QDIT (Bukharin & Zhao, 2023) | 43.3 | 14.5 | 81.1 | 54.4 | 45.2 | 32.7 | 45.2 |
| $k$-Center | 41.5 | 11.8 | 79.2 | 51.7 | 43.5 | 28.4 | 42.7 |
| $k$M-Closest | 42.1 | 14.2 | 80.4 | 54.9 | 44.6 | 31.2 | 44.6 |
| $k$M-Random | 43.2 | 15.4 | 81.0 | 55.5 | 44.8 | 31.2 | 45.2 |
| $k$MQ | 45.9 | 16.2 | **81.2** | 55.3 | 45.5 | 33.0 | 46.2 |
| Iterative $k$MQ | **46.1** | **18.4** | 80.1 | **56.0** | **46.3** | **34.3** | **46.9** |

Table 2: **Data selection performance of $k$MQ and baseline methods**. All methods sample 10K (5%) from the full WizardLM (196k) dataset. kMQ-$k$ denotes $k$-means-quality with $k$ clustering centroids. For both $k$M-Closest and $k$M-Random, we show the results of the optimal $k$.

## 4 RESULTS AND DISCUSSION

### 4.1 MAIN FINDINGS

Table 2 presents the performance of the proposed methods for instruction data selection compared to several baselines across various tasks. Our first observation is that by clustering data points using the $k$-means method and randomly sampling instances ($k$**M-Random** sampling) we already outperform random sampling and achieve comparable results to strong baselines: Deita and QDIT. This is significant because this sampling method is significantly more efficient than both Deita and QDIT and does not depend on costly LLMs for scoring. The success of this simple and efficient method highlights the impact of prioritizing diversity in sampling.

Next, we observe that by replacing the random selection step with the quality-based approach ($k$**MQ**) we can improve model performance on all downstream tasks. $k$MQ outperforms strong sampling approaches, Deita (Liu et al., 2023) and QDIT (Bukharin & Zhao, 2023), on all tasks. Next, we observe that the iterative sampling approach (**Iterative $k$MQ**), which leverages early training signals to refine the selected subset, outperforms all previous baselines on most tasks. This suggests that the iterative process of resampling and finetuning based on cluster performance can effectively identify and prioritize high-quality instruction data, leading to better task performance.

Overall, our findings highlight the impact of a diversity-focused sampling approach, which selects a compact yet representative subset of the data through clustering and weighted sampling from the clusters. We find that it is also crucial to consider a feedback loop from the finetuning model and understand how it perceives and learns from the data. By incorporating this feedback we ensure that the sampling process aligns with the model's learning behavior for optimal results.

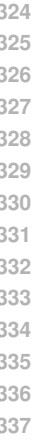

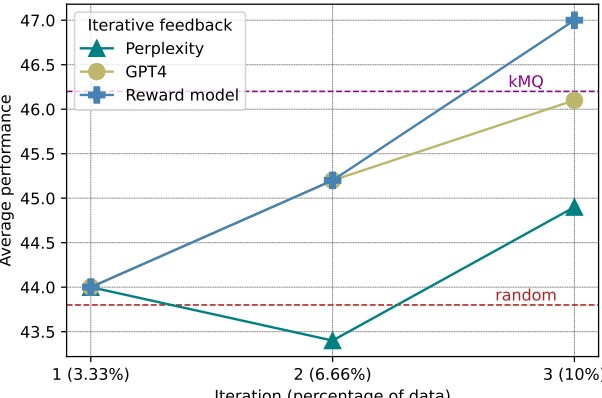

Figure 2: **Comparison of iterative selection approach using different sample-scoring methods**: perplexity, GPT-4, reward model. Note that both *random* and *kMQ* selection methods use 10% of data and train for three epochs. The iterative feedback runs are performed with the same budget at iteration 3, ensuring a fair comparison. Iterative sampling using a reward model achieves the best performance.

### 4.2 COMPARING DIFFERENT SCORING METHODS IN ITERATIVE FEEDBACK

To study the impact of how we score samples during training in our **iterative selection** approach, we compare three methods: calculating the perplexity score of generations, using GPT-4 to obtain a quality score, and using a reward model's[1] output. In Figure 2 we observe that all three variants effectively improve the average performance over random selection. It is important to note that during the first and second iterations, the iterative methods have been exposed to fewer data points compared to the random and $k$MQ baselines. It is only at the third iteration that all methods have had the opportunity to process an equal amount of data. While both perplexity-based and GPT-4-based scoring achieve similar performance to $k$MQ and improve over random sampling, the reward model variant largely outperforms a single-run $k$MQ. For this experiment, we arbitrarily selected an iteration value of 3, which can be modified in future experiments.

### 4.3 IMPACT OF NUMBER OF CLUSTERS

In $k$-means data selection, an important question is how to choose the appropriate value for the parameter $k$ (the number of clusters). Increasing the value of $k$ results in more fine-grained clusters and by ensuring that we sample from each cluster, we can increase the diversity of the selected subset. However, overly large values of $k$ would also inevitably create outlier clusters that consist entirely of low-quality, noisy data. Since we ensure each cluster is represented in the final selection, this results in noise being included. There is no one-size-fits-all answer, as the optimal $k$ depends on the characteristics of the pool of data. Exploring the optimal parameter value is costly, as it must be conducted with each new dataset. Here we use established heuristics in the clustering literature to guide this decision and study the correlation of these metrics with downstream performance of language models. Namely we investigate two methods:

**Elbow method** is a popular approach (Ahmed et al., 2020), where the objective value is plotted against different values of $k$. The goal is to identify the *elbow point*, where increasing $k$ yields diminishing returns in the performance.

**Silhouette Score** (Vardakas et al., 2024) provides another perspective by evaluating how well each data point fits within its assigned cluster (cohesion) compared to other clusters (separation), ranging from -1 (poor fit) to 1 (excellent fit). A high score indicates the object is similar to others in its cluster and dissimilar to those in neighboring clusters.

---

[1]We use `FsfairX-LLaMA3-RM-v0.1` (Xiong et al., 2024).

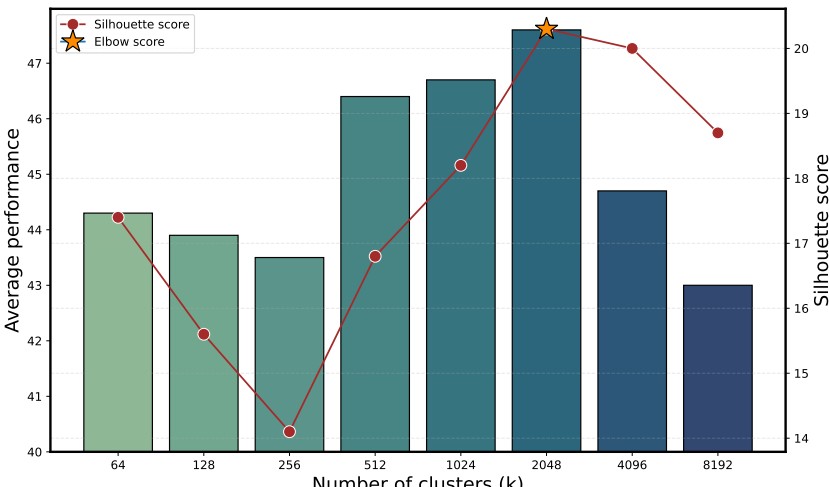

Figure 3: **Average performance on downstream tasks (bar plots) for different number of clusters** $k$. There is a correlation between downstream performance and both Silhouette and Elbow scores. The silhouette score is an efficient and effective proxy to estimate the number of clusters eliminating the need to explore the hyperparameter space.

Although both approaches for identifying the ideal number of clusters are frequently employed, the Silhouette score is generally preferred to the Elbow method in $k$-means clustering due to its clear interpretability, robustness to noise and outliers, and suitability for high-dimensional data. More importantly, the Elbow method is a post-hoc evaluation metric after the instruction tuning is done and is more expensive; while *Silhouette* score can be computed prior to any sampling and training and is very cheap.

We study how the choice of $k$ affects performance on downstream tasks and if we can identify an optimal $k$ based on the dataset's properties. To investigate this, we first fix our training pool (using WizardLM) and run a series of experiments with different numbers of clusters $k$. For each value $k$, we cluster the training candidates and sample from the clusters to create subsets of instruction data. We then finetune a model on each of these subsets. A full evaluation is conducted for every finetuned model (see detailed results in Appendix B).

Figure 3 provides a summary of the results, we reported the average performance over all tasks and observe that the average performance changes dramatically when we change the number of clusters. This is expected since we rely on the clustering step to group data points that are similar and distinct from other clusters. Remarkably, we observe a correlation between performance on downstream tasks and the Silhouette score. We find that the Silhouette score can be used to estimate the number of clusters required before performing the expensive pipeline of clustering, sampling, finetuning, and evalua-

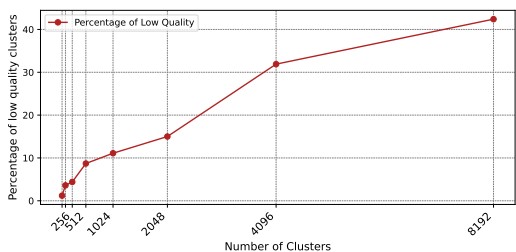

Figure 4: The percentage of clusters with an aggregated quality score below the threshold of 0.3.

tion. This estimation step enables us to adapt our approach efficiently to new datasets and collections, ensuring optimal performance and reducing computational costs associated with trial-and-error methods to find the best hyperparameters.

## 4.4 ANALYZING CLUSTER QUALITY

In our approaches, we rely on $k$-means clustering to ensure high diversity, but there is a risk that some clusters may consist solely of noise. To understand how this varies with different values of $k$, we use a reward model to evaluate the quality of each cluster with a score between 0 and 1. Figure 4 shows

the number of clusters with average quality scores below a certain threshold (0.3) for different values of $k$. We observe that by increasing the number of clusters, the percentage of the clusters dominated by low-quality data also increases. This increases the likelihood of sampling low-quality data when attempting to ensure that every cluster is represented in the final selection. In our iterative sampling approach, we adjust cluster weights during each training iteration and prevent noisy clusters from being over-represented in the sampled data.

|  | MMLU | GSM8K | HellaSwag | ARC | TruthfulQA | HumanEval | **Avg.** |
|---|---|---|---|---|---|---|---|
| Mistral-7B | | | | | | | |
| Random | 58.2 | 26.2 | 82.4 | 60.1 | 60.5 | 26.3 | 52.3 |
| $k$MQ | 59.1 | 31.0 | 83.3 | 60.2 | 64.7 | 28.4 | 54.5 |
| Iterative $k$MQ | 59.6 | 32.2 | 83.5 | 60.1 | 66.8 | 29.7 | **55.3** |
| Llama-3-8B | | | | | | | |
| Random | 65.1 | 38.4 | 83.3 | 60.6 | 55.1 | 56.7 | 59.9 |
| $k$MQ | 67.2 | 40.1 | 83.5 | 61.3 | 57.3 | 57.6 | **61.2** |
| Iterative $k$MQ | 66.0 | 36.7 | 83.3 | 61.0 | 56.4 | 54.2 | 59.6 |

Table 3: **Performance of our best sampling methods on downstream tasks for two base models:** Llama-3-8B and Mistral-7B. We sample 10K (5%) from WizardLM (196k). The selection is performed with Llama-2.

## 4.5 TRANSFERABILITY OF RESULTS

We conduct experiments with two additional base models, Mistral-7B and Llama-3 8B (Jiang et al., 2023; Dubey et al., 2024), to assess whether our findings generalize to other model families and more powerful models. Our results in Table 3 demonstrate that the effectiveness of iterative refinement remains valid for the Mistral-7B model, which exhibits more robust performance. However, the evaluation results for Llama-3 are mixed across different benchmarks. We observe improvements on average with $k$MQ sampling and a slight decrease in performance with iterative sampling especially in reasoning tasks. We hypothesize that Llama-2 differs from Mistral in its training data, model parameters, and training strategies. Consequently, using Llama-2 as a scorer reveals novel data points from which Mistral can benefit. However, Llama-3, a more advanced model than its predecessors with extended training as one of the primary distinctions, uncovers fewer new, valuable data points for further learning. This highlights that the quality scorer's effectiveness can vary, sometimes proving more beneficial and other times less so, depending on the base model for which we are sampling.

While the iterative refinement pipeline can select a dataset restricted to certain models, we do not view this as a limitation. The primary contribution of this work is to propose a function that takes a fixed dataset and model as input and outputs the most valuable subset for learning. This approach aligns with similar works (Ilyas et al., 2022; Thrush et al., 2024). Specifically, the task is to extract a subset of data that leverages early reward signals to enhance the targeted model's post-training performance.

## 5 RELATED WORK

**Data selection for LLMs.** Previous works on data selection can be broadly categorized into two key approaches: (1) removing undesirable examples, for instance, low-quality (Raffel et al., 2023; Marion et al., 2023), toxic (Raffel et al., 2023), or duplicated instances (Zhang et al., 2022; Abbas et al., 2023). (2) identifying the most optimal subset of data. While the definition of an optimal subset varies across different works, the shared goal is to use a small portion of the data while still maintaining strong performance. This subset selection approach has often been done by aiming for selecting high-quality instances through a proxy: manual curation (Zhou et al., 2023), selecting instances from human-authored datasets (Wang et al., 2024b), or hand-selecting datasets encouraging complexity and diversity (Ivison et al., 2023). More recently, a line of work has used language models to assess the quality of each data point and select the best ones. Xia et al. (2024) estimates data influence and performs a low-rank gradient similarity search using a gradient datastore. Liu et al.

(2023) scores instances using a combination of complexity and quality scores using an LLM and selects the final subset using diversity-based filtering. While individual sample quality is a crucial factor, prioritizing this local criterion can limit the diversity of the final selection. However, diversity in instances and tasks is essential for training high-performant models (Wei et al., 2021; Gudibande et al., 2023). Our work differs from these studies by examining what constitutes an optimal subset from a global perspective and by prioritizing representativeness. Closest to our work, Bukharin & Zhao (2023) emphasized quality by encoding all data points in the selection pool using an embedding model and selecting the final subset based on pairwise cosine similarity and a quality score from an LLM. In contrast, our approach offers a significantly more efficient method for subset selection, while also achieving improved performance. Our experiment covers multiple dimensions, including various base models, different encoding and scoring methods, and extensive ablation studies with recommendations for efficient parameter selection.

**Active learning and language models.** Active learning is based on the fundamental premise that "not all data is equal". This approach aims to identify the most informative data for pretraining or adapting language models for specific tasks or capabilities, as well as pinpointing the most valuable data for learning. Margatina et al. (2023) explored active learning for selecting in-context examples in few-shot learning, demonstrating that similar examples outperform uncertain or diverse in-context examples. Muldrew et al. (2024) proposed active preference learning, combining iterative data acquisition with a DPO (Direct Preference Optimization) loop to reduce the frequency of querying human annotators (Oracle). Their acquisition method relies on the model's entropy during generation. Our approach generalizes active instruction tuning (Kung et al., 2023) to instance-level data selection, allowing for the co-evolution of the LLMs and instruction data using an external reward signal.

## 6 CONCLUSION

In this paper, we present a novel approach to selecting a subset of data and optimizing the fine-tuning of language models. Our method involved a scalable sampling technique that maximizes diversity and efficiency in subset selection. Through our proposed $k$-means-quality ($k$MQ) algorithm and iterative selection process, we demonstrated significant performance improvements over strong baselines while maintaining a limited training budget. Our contributions include an efficient instruction selection algorithm, the release of our encoded instruction dataset, and a systematic analysis of our method's effectiveness across a range of tasks. Our method outperforms existing baselines, achieving up to 7% improvement in a wide range of evaluation tasks.

By addressing the challenge of optimal instruct data selection, our work paves the way for more efficient and effective finetuning of language models, making them more accessible and affordable for deployment, especially in resource-constrained settings. We believe that our findings will contribute significantly to the ongoing research in language model optimization and their real-world applications.

## 7 LIMITATIONS AND FUTURE WORK

While our proposed method has shown promising results, there are a few limitations to consider. Our evaluation focused on a specific set of tasks, and future work can aim to validate our method's effectiveness across a broader range of language models and tasks, including data selection in the pre-training stage and alignment (Yu et al., 2024; Muldrew et al., 2024). Furthermore, our iterative selection process relies on early training signals, and we only presented this as a pilot study to encourage further research. Future work could explore alternative model feedback mechanisms to refine the selected instruction data subsets, especially in mitigating the potential for reward hacking in the iterative refinement process (Pan et al., 2024).

Finally, while we considered diversity and difficulty crucial factors, other characteristics of instruction data could be explored to enhance the finetuning process further. Addressing these limitations and extending this research will contribute to more robust and adaptable language models, capable of excelling in a wide range of real-world applications.

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

APPENDIX

# A  TRAINING DETAILS

## A.1  HYPERPARAMETERS

For supervised fine-tuning, our training hyperparameters are presented in table 4.

| Parameter | Value |
|---|---|
| Precision | BFloat16 |
| Epochs | 3 |
| Selected Portion | 10% |
| Gradient Accumulation Step | 8 |
| Batch Size | 64 |
| Max Seq. Length | 4096 |
| K-means Random Seed | 42 |

Table 4: Our training hyperparameters.

## A.2  COMPUTATIONAL COST

We also utilised Deepspeed-Zero3 (Rasley et al., 2020) for better efficiency training. Models are finetuned with combination of TPU and GPU. For TPU, we used the code provided by `young-geng/EasyLM`[2] and done with TPUv3-32 nodes. For GPU, 2 A100-80GB are used across the fine-tuning.

# B  IMPACT OF NUMBER OF CLUSTERS

| Method | MMLU | GSM8K | HellaSwag | ARC | TruthfulQA | HumanEval | Avg. | Silhouette Score |
|---|---|---|---|---|---|---|---|---|
| $k$MQ-64 | 43.1 | 13.9 | 80.2 | 54.3 | 44.8 | 29.5 | 44.3 | 17.4 |
| $k$MQ-128 | 43.4 | 12.8 | 79.9 | 54.1 | 45.0 | 28.4 | 43.9 | 15.6 |
| $k$MQ-256 | 42.3 | 13.1 | 80.0 | 53.2 | 44.3 | 28.1 | 43.5 | 14.1 |
| $k$MQ-512 | **46.4** | 17.0 | 81.2 | 55.3 | **45.5** | 33.0 | 46.4 | 16.8 |
| $k$MQ-1024 | 45.6 | 17.8 | 81.6 | **55.5** | 44.9 | 34.1 | 46.6 | 18.2 |
| $k$MQ-2048 | 46.0 | **20.2** | **82.1** | **55.5** | 45.0 | **37.2** | **47.7** | **20.3** |
| $k$MQ-4096 | 44.2 | 15.2 | 79.1 | 54.3 | 42.0 | 33.2 | 44.7 | 20.0 |
| $k$MQ-8192 | 44.1 | 13.6 | 78.9 | 54.2 | 41.6 | 31.8 | 43.0 | 18.7 |

Table 5: Performance of models trained on different number of data clusters $k$. We sample 10K (5%) for each experiment. Silhouette score correlates with downstream tasks and is an efficient proxy for estimating the number of clusters before sampling.

# C  SCORER DETAILS

For perplexity, we pass the $x_i \oplus y_{gen}$ and $x_i \oplus y_{gold}$ to the model to compute the perplexity scores. The scorer with Perplexity is as follows:

$$\mathcal{S}(x_i, y_{\text{gen}}, y_{\text{gold}}) = -\log(\frac{PPL(x_i \oplus y_{\text{gen}})}{PPL(x_i \oplus y_{\text{gold}})}) \tag{7}$$

For GPT-4 direct scoring, we give the two completions to GPT-4 and ask it to give a rating between 1 and 5. We use the template as shown in Figure 5 to prompt GPT-4 for being the LLM-as-a-judge and

---

[2]young-geng/EasyLM

|  | MMLU | GSM8K | HellaSwag | ARC | TruthfulQA | HumanEval | **Avg.** |
|---|---|---|---|---|---|---|---|
| Random | 38.2 | 9.1 | 79.1 | 51.3 | 41.1 | 20.5 | 39.9 |
| Deita | 39.4 | 10.7 | 79.4 | 51.2 | 41.7 | 22.9 | 40.9 |
| QDIT | 38.7 | 11.3 | **79.8** | **51.6** | 42.6 | 25.6 | 41.6 |
| $k$-Center | 37.3 | 8.1 | 79.0 | 50.7 | 41.0 | 12.8 | 38.2 |
| $k$M-Closest | 40.1 | 10.3 | 79.3 | 51.2 | 42.5 | 24.3 | 41.3 |
| $k$M-Random | 39.6 | 11.4 | 79.1 | 51.2 | 42.8 | 25.1 | 41.5 |
| $k$MQ-64 | **41.3** | **12.6** | 79.7 | 51.1 | **43.4** | 25.3 | **42.2** |
| $k$MQ-256 | 39.5 | 12.3 | 79.1 | 51.0 | 42.7 | **26.0** | 41.8 |
| $k$MQ-1024 | 37.3 | 11.2 | 78.6 | 51.2 | 41.5 | 22.3 | 40.4 |

Table 6: Additional experiments on Alpaca dataset (52k). We sample 5K (10%) for each experiment. kMQ-$k$ denotes $k$-means-quality with $k$ clustering centroids. For both $k$M-Closest and $k$M-Random, we show the results of the optimal $k$ among all choices of $k$.

|  | Size | MMLU | GSM8K | HellaSwag | ARC | TruthfulQA | HumanEval | **Avg.** |
|---|---|---|---|---|---|---|---|---|
| Random | 10k | 42.4 | 13.3 | 79.9 | 53.6 | 44.8 | 28.5 | 43.8 |
| Iter-1 | 3.3k | 44.3 | 14.5 | 79.7 | 54.5 | 44.7 | 26.1 | 44.0 |
| PPL Iter-2 | 6.7K | 41.8 | 13.4 | 80.1 | 52.4 | 44.2 | 27.8 | 43.4 |
| PPL Iter-3 | 10K | 43.9 | 15.6 | 79.9 | 55.1 | 45.6 | 30.4 | 44.9 |
| GPT Iter-2 | 6.7K | 44.6 | 14.8 | 79.6 | 54.2 | 45.8 | 32.1 | 45.2 |
| GPT Iter-3 | 10K | 45.4 | 16.9 | **80.2** | 55.0 | 45.7 | **34.5** | 46.1 |
| RM Iter-2 | 6.7K | 44.7 | 15.8 | 80.1 | 54.7 | 45.2 | 30.8 | 45.2 |
| RM Iter-3 | 10K | **46.1** | **18.4** | 80.1 | **56.0** | **46.3** | 34.3 | **47.0** |

Table 7: Performance of our best iterative sampling method (using a reward model) on different test sets. The training pool is WizardLM (196k). We plot the results in Figure 2. Best scores are bold. Second bests are underlined.

by replacing the reward scoring ($R$) by the GPT score in Equation (4). The template is inspired by Zheng et al. (2023). For the reward model, we use an off-the-shelf model based on Llama-3[3].

---

**Prompt Template for Judgment Annotation**

Please act as an impartial judge and evaluate the quality of the response provided by an AI assistant to the user question displayed below. Your evaluation should consider factors such as the helpfulness, relevance, accuracy, depth, creativity, and level of detail of the response. Begin your evaluation by providing a short explanation. Be as objective as possible. After providing your explanation, please rate the response on a scale of 1 to 10 by strictly following this format: "[[rating]]", for example: "Rating: [[5]]".

[[Instruction]]
${instruction}

[[Response]]
${response}

---

Figure 5: Prompt template for requesting a response evaluation from GPT-4-turbo, where variables ${instruction} and ${response} are replaced with examples in our dataset.

---
[3]FsfairX-LLaMA3-RM-v0.1

