# OpenReview forum: "Diversify and Conquer: Diversity-Centric Data Selection with Iterative Refinement"
_ICLR.cc/2025/Conference — Submitted to ICLR 2025_

### Official Review · Reviewer_Jz4C · 2024-10-29

**Soundness:** 3
**Presentation:** 2
**Contribution:** 2
**Rating:** 3
**Confidence:** 4

**Summary:**

Finetuning LLMs on instruction data is an important step to improve instruction following capabilities. This paper attempts to address how to determine the optimal subset of data for effective training. Unlike previous works that focused on local criteria (e.g., data quality), this paper claims that data diversity is more critical for effective training. This paper utilizes k-means clustering and proposes an iterative refinement method inspired by active learning techniques to resample instances from clusters. Experiments show that the proposed method achieves a 7% increase over the random selection and a 3.8% improvement over state-of-the-art sampling methods.

**Strengths:**

1. Even though the claim that data diversity is critical is well-known [1,2], the motivation of this paper is clear.
2. The proposed method is sound. Using active learning is a straightforward idea to select the optimum subset of data.
3. This paper is well-written and easy to follow. The released code and the data may facilitate reproducibility and future research.

Overall, this paper attempts to address an important question and proposes an effective method that achieves better performance than the compared methods.

[1] LIMA: Less Is More for Alignment

[2] Data Diversity Matters for Robust Instruction Tuning

**Weaknesses:**

1. Baselines are weak: Even though the author attempts to demonstrate that data diversity is more important than data quality, the quality-based data selection method chosen by the authors is insufficient to support their claims, as they have selected only one method based on quality. (i.e., Deita,). There are still lots of published papers focusing on data quality [1,2,3]. Meanwhile, the improved performance compared with Random Selection is also not convincing.
2. Evaluation is weak: Even though the paper claims that finetuning LLMs on instruction data is an important step to improve instruction following capabilities, there is no evaluation of the quality of the generated response (e.g., MT-Bench [4]). The evaluation in the paper mainly focuses on understanding tasks instead of instruction-following tasks.

[1] From Quantity to Quality: Boosting {LLM} Performance with Self-Guided Data Selection for Instruction Tuning

[2] Clustering and Ranking: Diversity-preserved Instruction Selection through Expert-aligned Quality Estimation

[3] LESS: Selecting Influential Data for Targeted Instruction Tuning

[4] Judging LLM-as-a-judge with MT-bench and chatbot arena

**Questions:**

None.

---

> ### Author Response · Authors · 2024-11-22
>
> ​​We appreciate the reviewer’s close reading of the paper and would like to thank the reviewer for “liking the work” and appreciating “Experiments and ablations are conducted over several datasets”. We respond to the reviewer’s concerns below.
>
> 1. Thank you for highlighting the related work. Data selection is indeed a broad and long-standing research area. Comparing against every new data selection method would require extensive experiments for sampling and evaluation, which is not scalable. Therefore, we based our baseline selection on two criteria: high-performing state-of-the-art approaches and similarity in methodology and execution cost.
>
> Regarding the specific references:
>
> Reference [2]: This work was published just last week, after the conference submission deadline.
>
> Reference [1]: Their approach involves training a pre-experienced model, making it significantly more computationally expensive than ours. Additionally, determining the optimal number of "cherry" samples is unclear, and exploring such a costly parameter space is not feasible.
>
> Reference [3]: This method relies on constructing a gradient datastore, which is computationally intensive.
> For both references [1] and [3], a direct comparison would not be fair due to their reliance on computationally expensive offline storage methods.
>
> 2. Thanks for the valuable feedback. We have run additional experiments to include evaluation results on two main benchmarks: MT-Bench and AlpacaEval.
>
> | Trained in WizardLM | MT-Bench | AlpacaEval |
> |----------|:-------------:|------:|
> | Random | 7.3 | 71.0 |
> | Deita | 7.5 | 73.3 |
> | QDIT | 7.5 | 75.7 |
> | kMQ | 7.4 | 76.2 |
> | Iterative kMQ | 7.8 | 80.1 |
>
>
> Consistent with the existing results in the paper, we observe that both kMQ and iterative kMQ outperform baselines on both benchmarks, with AlpacaEval highlighting a larger difference.

---

### Official Review · Reviewer_wtSp · 2024-10-30

**Soundness:** 3
**Presentation:** 2
**Contribution:** 2
**Rating:** 3
**Confidence:** 4

**Summary:**

This paper introduces use of clustering for data subset selection using clustering followed by quality score sampling for efficient instruction tuning. Paper proposes multiple variants of the clustering based method, and how to do unsupervised tuning of the cluster center hyperparameter. I also appreciate the use of reward model as a score function, which goes beyond the classic loss based score such as perplexity. Experiments and ablations are conducted over several datasets, including transferability, showing robustness of the proposed approach.

**Strengths:**

Paper proposes on how to do unsupervised tuning of the cluster center hyperparameter. I also appreciate the use of reward model as a score function, which goes beyond the classic loss based score such as perplexity. Experiments and ablations are conducted over several datasets. As far as this study is concerned for clustering based solutions, I like this work.

**Weaknesses:**

A major weakness of this work is lack of literature review. For example, use case of diversity and some sort of quality score has been used in Machine Learning for a long time, from active learning [1], to curriculum learning [2, 6] (also see common followups to [6] which are on the similar lines, such as [7]), to more recent LLM instruction tuning based works [3, 4, 5]. However, paper fails to acknowledge these works except [3]. Submodular functions have been used for the purposes of inducing diversity for a very long time, which also includes determinantal point process. However, no discussion on submodularity has been provided in the related works.

I believe [5] has done somewhat detailed study on submodular functions, its usecase with quality scores (uncertainty, confidence) and I think the perplexity based score function may have some correlation between confidence based score function.

 To summarize, I think this is a good paper when it comes to the clustering based methods, but it needs to benchmark against the submodular methods as well.

References -

[1] Submodularity in Data Subset Selection and Active Learning
[2] Curriculum Learning by Dynamic Instance Hardness
[3] An Experimental Design Framework for Label-Efficient Supervised Finetuning of Large Language Models
[4] Diversity Measurement and Subset Selection for Instruction Tuning Datasets
[5] SmallToLarge (S2L): Scalable Data Selection for Fine-tuning Large Language Models by Summarizing Training Trajectories of Small Models
[6] Coresets for Data-efficient Training of Machine Learning Models
[7] Towards Sustainable Learning: Coresets for Data-efficient Deep Learning

**Questions:**

- For k-means, why is k-means++ initialization not used, instead of using 42 as random seed?
- I am interested in seeing some run-time analysis, as usually the size of the instruction tuning dataset is large.
- Can the authors try to run experiments using facility location alone, and then using that instead of clustering? It may be important to note that different choice of kernel in facility location can result in different performance (Cosine v/s RBF as done in [3]). Note that one has to do tuning, as they've also suggested a way to tune the resulting submodular function.
- Can authors also try to compare against DPP as done in [4] ?

---

### Official Review · Reviewer_wrrm · 2024-11-03

**Soundness:** 3
**Presentation:** 3
**Contribution:** 2
**Rating:** 5
**Confidence:** 3

**Summary:**

This study addresses the challenge of selecting optimal data subsets for fine-tuning LLMs on instruction datasets by focusing on data diversity rather than solely on local quality criteria. Utilizing a k-means clustering method, the approach aims to represent the entire dataset effectively, with an iterative refinement inspired by active learning to continuously adjust sampling weights and filter out low-quality data clusters. The experiemnts illustrate that this diversity-first approach leads to an improvement over random sampling showcasing the value of diverse data selection for enhancing LLM performance across various tasks.

**Strengths:**

1. The proposed data selection method achives a better downstream performance than evaluated baselines, which showcases the importance of diversity in data selection.

2. The downstream analysis datasets are extensive.

**Weaknesses:**

1. I think the baseline comparison is not complete. For example, there is a paper "One-Shot Learning as Instruction Data Prospector for Large Language Models" published on ACL 2024 which is higly related to this topic but not been included as baseline in this paper.

2. The experiment still cannot persuade me that the diversity is the most imporant factor in selecting instruciton data. In Table 3, the proposed sampling method seems marginally better than the Random selection. In some datasets, it is even worse than Random selection, such as HumanEval on Llama-3.

**Questions:**

1. Please refer to Weaknesses.

---

> ### Author Response · Authors · 2024-11-22
>
> ​​We appreciate the reviewer’s close reading of the paper and agree that our work provides certain insights. We respond to the reviewer’s concerns below.
> > I think the baseline comparison is not complete. For example, there is a paper "One-Shot Learning as Instruction Data Prospector for Large Language Models" published on ACL 2024 which is higly related to this topic but not been included as baseline in this paper.
>
> We thank the reviewer for pointing out paper [1] as a highly related work and we have included it as the related work in the paper; however, we would like to emphasize our approach is significantly different from the proposed method. Our proposal mainly focuses on how diversity plays a major role in the field of data selection, and we achieved this in an unsupervised way using k-means clustering. The Nugget method in the proposed paper requires prior knowledge about the task in the dataset and matches each instance to the task, which increases the requirements for dataset selection. Additionally, we covered more recent baselines, including DEITA and QDIT, which use better quality estimators (DEITA uses a model trained for scoring distilled by ChatGPT, whereas QDIT directly utilises a reward model for quality measurement).
>
> > The experiment still cannot persuade me that the diversity is the most imporant factor in selecting instruciton data. In Table 3, the proposed sampling method seems marginally better than the Random selection. In some datasets, it is even worse than Random selection, such as HumanEval on Llama-3.
>
> Regarding the drop in “Iterative kMQ” performance in Table 3, we do acknowledge this in the paper and discuss it between lines 453 and 455. We identify the drop in performance as being due to the significant difference between the pre-training corpora of Llama-2 and Llama-3 (2T vs. 15T tokens). However, the kMQ method still outperforms the Random baseline for Llama-3 models, which shows the effectiveness of our proposed method. As pointed out between lines 467 and 472, we don’t see this as a disadvantage of our iterative method since there always exists a trade-off between finding the most optimal subset for data selection for specific models compared with a general dataset that fits every dataset.
>
>
> We hope our discussion answers the reviewer’s question. Please let us know if there are further concerns that we could help with.

---

### Official Review · Reviewer_PCKb · 2024-11-04

**Soundness:** 1
**Presentation:** 2
**Contribution:** 1
**Rating:** 1
**Confidence:** 4

**Summary:**

The authors propose a data subset selection strategy that considers both diversity and quality to select samples for finetuning LLM's. Specifically, the strategy clusters the points and selects samples with high quality/hardness scores from each cluster.

**Strengths:**

The paper is easy to understand and well-written, and the problem of selecting diverse and high quality samples for instruction tuning is important. However, I have several concerns regarding the design choices of the methodology and the experiments.

**Weaknesses:**

## Weaknesses
- **[Major]** The key sampling strategy used in this work is a reincarnation of strategies that have been used for a while in data pruning and active learning. For example, [1] (work not cited in this paper) first clusters the unlabeled set and chooses the samples with the highest uncertainty from each cluster in a round-robin fashion though it is very straightforward to substitute uncertainty with quality.
- **[Major]** There are many missing baselines in the experiments. Many active learning and subset selection strategies can be used for this setting from the core ML community [2,3,4] many of which have been applied to the problem of supervised finetuning [5]. There are also subset selection works specifically for supervised finetuning that are missing, for example [6]. At least a few of these should be included as baselines in this work, and discussion of all of these works should be included in the related works section.
- **[Major]** Additional ablations are necessary. What happens if we only use quality without any diversity prefiltering?
- **[Major]** In most of the experiments, evaluation is only done on a single subset size. Evaluation on multiple subset sizes is fairly standard in data selection literature even at large scales [5,6], and should be included here as well.
- **[Major]** The strategy first clusters the points and then samples based on quality. However, this does not guarantee diversity in the final subset. Using quality based sampling in the second stage can still induce redundancies **within a cluster**. There is nothing to prevent two identical high quality samples that belong to the same cluster from being selected. Filtering based on some score before applying diversity sampling prevents this (this has been done in [7]).
- **[Minor]** Why is the sampling strategy randomized? From my understanding, sampling is performed only once for each iteration and scores must be computed for each sample to compute its probability so there doesn't seem to be any benefit for this in terms of computational cost.
- **[Minor]** In Algorithm 1, why is $\mathcal{D}'$ grown incrementally? In other words, a new set of points is sampled from $\mathcal{D}/\mathcal{D'}$ and added to $\mathcal{D}'$ at each round, but shouldn't all of $\mathcal{D'}$ be resampled? It seems as though points that the LLM has already been trained on will be easy and assigned a low score based on Eq. 4.
- **[Minor]** This claim "The success of this simple and efficient method highlights the impact of prioritizing diversity in sampling." does not make sense given that the baseline method QDIT also is a diversity-based selection approach.

I know I have listed several points and I understand that it may not be possible to address all of them in the short rebuttal period, so I would encourage the authors to prioritize addressing the **[Major]** points.

## References
- [1] Batch Active Learning at Scale (https://arxiv.org/pdf/2107.14263)
- [2] Deep Batch Active Learning by Diverse, Uncertain Gradient Lower Bounds (https://arxiv.org/abs/1906.03671)
- [3] D2 Pruning: Message Passing for Balancing Diversity and Difficulty in Data Pruning (https://arxiv.org/abs/2310.07931)
- [4] Active Learning for Convolutional Neural Networks: A Core-Set Approach (https://arxiv.org/abs/1708.00489)
- [5] An Experimental Design Framework for Label-Efficient Supervised Finetuning of Large Language Models (https://arxiv.org/abs/2401.06692)
- [6] Diversity Measurement and Subset Selection for Instruction Tuning Datasets (https://arxiv.org/pdf/2402.02318)
- [7] Submodularity in Data Subset Selection and Active Learning (https://proceedings.mlr.press/v37/wei15.html)

**Questions:**

See weaknesses

---

> ### Author Response · Authors · 2024-11-22
> **Rebuttal to the reviewer**
>
> We appreciate the reviewer’s suggestions, which helped us to improve our paper, but we do want to emphasize that there is some misunderstanding from the reviewer, which might affect the review of our paper. We respond to the reviewer’s concerns correspondingly as follows:
>
> > [Major] The key sampling strategy used in this work is a reincarnation of strategies that have been used for a while in data pruning and active learning. For example, [1] (work not cited in this paper) first clusters the unlabeled set and chooses the samples with the highest uncertainty from each cluster in a round-robin fashion though it is very straightforward to substitute uncertainty with quality.
>
> We appreciate the reviewer pointing out comprehensive references; we have included some in the related work section of the modified preprint version. However, it’s important to note that some papers aren’t relevant to our work, and it’s impossible to run every baseline (especially when previous works [1][2] also didn’t cover them). For instance, [3] requires significant computation on the gradient, which is nearly impossible to apply in current large-scale models, as it becomes exhaustive and intractable. Additionally, [4] mainly works for CNN networks, and it’s nearly impossible to transfer it to the current transformer architecture. We hope the reviewer could reconsider the scope of this paper, as we are mainly focusing on proposing iterative instruction data selection, which is currently under-explored.
>
> > [Major] There are many missing baselines in the experiments. Many active learning and subset selection strategies can be used for this setting from the core ML community [2,3,4] many of which have been applied to the problem of supervised finetuning [5]. There are also subset selection works specifically for supervised finetuning that are missing, for example [6]. At least a few of these should be included as baselines in this work, and discussion of all of these works should be included in the related works section.
>
> We appreciate the reviewer pointing out comprehensive references; we have included some in the related work section of the modified preprint version. However, it’s important to note that some papers aren’t relevant to our work, and it’s impossible to run every baseline (especially when previous works [1][2] also didn’t cover them). For instance, [3] requires significant computation on the gradient, which is nearly impossible to apply in current large-scale models, as it becomes exhaustive and intractable. Additionally, [4] mainly works for CNN networks, and it’s nearly impossible to transfer it to the current transformer architecture. We hope the reviewer could reconsider the scope of this paper, as we are mainly focusing on proposing iterative instruction data selection, which is currently under-explored.
>
> > [Major] Additional ablations are necessary. What happens if we only use quality without any diversity prefiltering?
>
> | Model                        | MMLU | GSM8K | HellaSwag | ARC  | TruthfulQA | HumanEval | Average |
> |------------------------------|------|-------|-----------|------|------------|-----------|---------|
> | kMQ (w/o diversity prefiltering) | 43.2 | 14.3  | 81.0      | 54.9 | 45.5       | 23.4      | 43.7    |
> | kMQ (Ours)                   | 45.9 | 16.2  | 81.2      | 55.3 | 45.5       | 33.0      | 46.2    |
> | Iterative kMQ                | 46.1 | 18.4  | 80.1      | 56.0 | 46.3       | 34.3      | 46.9    |
>
> > [Major] The strategy first clusters the points and then samples based on quality. However, this does not guarantee diversity in the final subset. Using quality-based sampling in the second stage can still induce redundancies within a cluster. There is nothing to prevent two identical high-quality samples that belong to the same cluster from being selected. Filtering based on some score before applying diversity sampling prevents this (this has been done in [7]).
>
> We appreciate the reviewer for pointing out this problem; however, we view it more as a deduplication method, as proposed in DEITA, which would only complicate our methods by reapplying the scoring base within a cluster. Although we expect minor improvements from using this, we would like to re-emphasize our contribution: we identify k-means as still an effective method and propose an iterative selection method, which we believe is underexplored for active instruction tuning. We have included this paper as part of the future work in the modified preprint version.
>
> References:
>
> [1] Data Diversity Matters for Robust Instruction Tuning (https://arxiv.org/abs/2311.14736)
> [2] What Makes Good Data for Alignment? A Comprehensive Study of Automatic Data Selection in Instruction Tuning (https://arxiv.org/abs/2311.14736)
> [3] Deep Batch Active Learning by Diverse, Uncertain Gradient Lower Bounds (https://arxiv.org/abs/1906.03671)
> [4] Active Learning for Convolutional Neural Networks: A Core-Set Approach (https://arxiv.org/abs/1708.00489)

---

> ### Author Response · Authors · 2024-11-22
> **Rebuttal [2/2]**
>
> > In Algorithm 1, why is D′ grown incrementally? In other words, a new set of points is sampled from D/D′ and added to D′ at each round, but shouldn't all of D′ be resampled? It seems as though points that the LLM has already been trained on will be easy and assigned a low score based on Eq. 4.
>
> This reviewer may have misunderstood our method. Here, our LLM is re-trained on the new D when D′ is added, instead of being continuously trained with the new data points. Therefore, it won’t result in the points that the LLM has already been trained on being assigned a low score. Regarding the question of why not all of D′ is resampled, this is a trade-off between diversity and quality. If D′ is resampled, then some clusters will be assigned higher scores and converge to certain clusters. Although this could improve the quality of the data selected, it is not optimal as it reduces the diversity of selection. Our results in Table 2 show that diversity is necessary.
>
> > **This claim "The success of this simple and efficient method highlights the impact of prioritizing diversity in sampling." does not make sense given that the baseline method QDIT also is a diversity-based selection approach.**
>
> For both QDIT and DEITA papers, they mentioned their methods as “diversity-aware selection.” However, their diversity method is mainly used for deduplication instead of diversity sampling like ours, with k-means methods (Eq 1 in QDIT [1] and Section 2.5 in DEITA). However, we appreciate the reviewer’s comments and explain this more clearly in the modified preprint.
>
> References:
>
> [1] Data Diversity Matters for Robust Instruction Tuning (https://arxiv.org/abs/2311.14736)
>
> [2] What Makes Good Data for Alignment? A Comprehensive Study of Automatic Data Selection in Instruction Tuning (https://arxiv.org/abs/2311.14736)
>
> [3] Deep Batch Active Learning by Diverse, Uncertain Gradient Lower Bounds (https://arxiv.org/abs/1906.03671)
>
> [4] Active Learning for Convolutional Neural Networks: A Core-Set Approach (https://arxiv.org/abs/1708.00489)

---

> ### Comment · Reviewer_PCKb · 2024-11-26
> **Response to Authors**
>
> I appreciate the author's response and additional experiments. However, I feel that most of my concerns are not addressed.
>
> > However, it’s important to note that some papers aren’t relevant to our work, and it’s impossible to run every baseline
>
> I do believe that the baselines still can be significantly improved. Given that diversity is known to be important in instruction tuning tasks [5] and the method proposed in this work is not new, the proposed method should achieve significant gains over the currently popular approaches in order for the work to have a meaningful contribution.
>
> > previous works [1][2] also didn’t cover them
>
> [1,2] do not compare against some of them as they predate a lot of the approaches that are popular now ([2] is from 2019).
>
> > [3] requires significant computation on the gradient, which is nearly impossible to apply in current large-scale models,
>
> Instead of using the full gradient, gradient approximations (for example the gradients w.r.t the last few layers) can be used to make [3] tractable.
>
> > [4] mainly works for CNN networks, and it’s nearly impossible to transfer it to the current transformer architecture.
>
> The title of [4] is misleading, but the algorithm can easily be extended to LLM's as done by [5] ([4] is not specific to CNN's, the theorems in that paper only hold for high capacity models like CNN's which can achieve 0 loss on the training set).
>
> >  However, their diversity method is mainly used for deduplication instead of diversity sampling
>
> Can the authors elaborate on what the distinction between diversity sampling and deduplication is? This is not clear to me.

---

### Meta-Review · Area_Chair_i5ES · 2024-12-12

**Metareview:**

This paper proposes considering both quality and diversity in the data selection process, to select instructional data better for follow-up tasks. The motivation is overall clear. Besides, the technical details of the proposed method are easy to follow.

This paper has some flaws that cannot be addressed by slight modifications. First, it lacks sufficient comparison with previous work in terms of content and ideas, which makes the novelty and its technical contribution unclear. Second, the comparison method is not comprehensive enough. The experimental results are not convincing enough. Third, although the technical details of the paper are easy to follow, it does not provide an intuitive and convincing explanation of why the proposed method actually works.

AC carefully reviewed the paper, read the reviewers’ comments, the authors’ responses, and organized the discussion. The current form needs further polishes to reach acceptance standards. Therefore, the final recommendation is "reject".

**Additional Comments On Reviewer Discussion:**

During the reviewing process, four reviewers propose their concerns from the aspects of the idea, technical details, experimental designs, and empirical results. The author feedback is provided to address them. However, it cannot handle the raised concerns well. A series of concerns remain, especially in experimental setups and technical novelty. Also, the authors did not further respond to or address the issues raised by the reviewer during the rebuttal process. In internal discussions, reviewers also stated that their concerns were not adequately addressed. Therefore, based on the above, a rejection recommendation is made.

---

### Decision · Program_Chairs · 2025-01-22

Reject